# Improving Complex Reasoning with Dynamic Prompt Corruption: A soft prompt Optimization Approach

**Sinan Fan**[1,2*]**, Liang Xie**[3,4†]**, Chen Shen**[4†‡]**, Ge Teng**[4]**, Xiaosong Yuan**[4]**, Xiaofeng Zhang**[4]
**Chenxi Huang**[4] **Wenxiao Wang**[1,4]**, Xiaofei He**[1]**, Jieping Ye**[4]
[1]Zhejiang University  [2]Hangzhou YunQi Academy of Engineering
[3]College of Computer Science and Technology, Zhejiang University of Technology
[4]Alibaba Cloud Computing

## Abstract

Prompt-tuning (PT) for large language models (LLMs) can facilitate the performance on various conventional NLP tasks with significantly fewer trainable parameters. However, our investigation reveals that PT provides limited improvement and may even degrade the primitive performance of LLMs on complex reasoning tasks. Such a phenomenon suggests that soft prompts can positively impact certain instances while negatively affecting others, particularly during the later phases of reasoning. To address these challenges, We first identify an information accumulation within the soft prompts. Through detailed analysis, we demonstrate that this phenomenon is often accompanied by erroneous information flow patterns in the deeper layers of the model, which ultimately lead to incorrect reasoning outcomes. we propose a novel method called **D**ynamic **P**rompt **C**orruption (DPC) to take better advantage of soft prompts in complex reasoning tasks, which dynamically adjusts the influence of soft prompts based on their impact on the reasoning process. Specifically, DPC consists of two stages: Dynamic Trigger and Dynamic Corruption. First, Dynamic Trigger measures the impact of soft prompts, identifying whether beneficial or detrimental. Then, Dynamic Corruption mitigates the negative effects of soft prompts by selectively masking key tokens that interfere with the reasoning process. We validate the proposed approach through extensive experiments on various LLMs and reasoning tasks, including GSM8K, MATH, and AQuA. Experimental results demonstrate that DPC can consistently enhance the performance of PT, achieving 4%-8% accuracy gains compared to vanilla prompt tuning, highlighting the effectiveness of our approach and its potential to enhance complex reasoning in LLMs.

## 1 Introduction

Prompt Tuning (PT) (Lester et al., 2021a) as a promising parameter-efficient fine-tuning (PEFT) approach demands very few trainable continuous prompt vectors added to the input, achieving competitive performance compared to the full-parameter fine-tuning (Qin et al., 2021; Ding et al., 2023a). Previous efforts have demonstrated the superiority of PT by generating high-quality soft prompts (Shi & Lipani, 2024; Yang et al., 2023; Qin et al., 2024). Nonetheless, soft prompting underperforms the foundation model on complex reasoning tasks. With a comprehensive investigation, we discover that vanilla prompt tuning provides little performance improvement on complex reasoning tasks and may even lead to negative performance outcomes.

From the perspective of individual instances, soft prompts may have positive effects for some instances while potentially posing a negative impact on others. Figure 1 illustrates an example from the GSM8K (Cobbe et al., 2021): the model can answer the question correctly without a soft prompt, while a soft prompt guides the LLM to an incorrect answer through some intermediate reasoning steps. Therefore, it is crucial to determine whether the effect of soft prompts on reasoning is positive or negative. However, understanding why certain reasoning succeeds while others fail is difficult.

---

*Work done during an internship at Hangzhou YunQi Academy of Engineering and Alibaba Cloud Computing.

†Corresponding Authors: Liang Xie <lilydedbb@gmail.com>, Chen Shen <jason.sc@alibaba-inc.com>.

‡Project Lead.

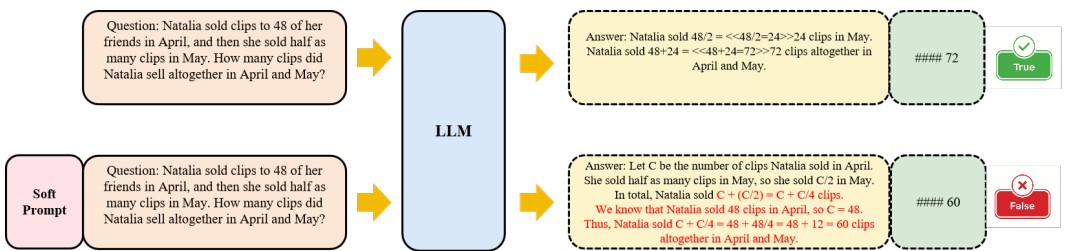

Figure 1: Input the same question to guide the LLM to answer it. The model was originally able to provide the correct answer, but after adding the soft prompts, it produced an error in reasoning.

To meet the challenge, we intend to explore the mechanisms of soft prompts on the Chain-of-Thought (CoT) (Wei et al., 2022) through a lens of information flow. Neuron saliency score analysis is an important tool for analyzing the information flow during LLM inference (Dai et al., 2022; Wang et al., 2023a), allowing to probe the models' reasoning mechanisms. After a comprehensive investigation across several LLMs and tasks, we find: *soft prompts lead to reaching the right answer when the latter reasoning steps put more emphasis on former tokens compared to soft tokens. In contrast, when the soft prompt significantly impacts the latter reasoning steps, the model may reach an incorrect answer.* This phenomenon revealed by neuron saliency score analysis is similar to the thinking process of human beings to some extent: humans always recall the background knowledge related to the question and then answer it. As a carrier of task-related knowledge (Xiao et al., 2024) (Yu et al., 2024), soft prompts can provide beneficial hints to help models understand the question and generate the initial reasoning steps, and then use the question and generated steps to complete the answer. However, if the remaining reasoning steps still attend to the hints, i.e., soft prompts, it may conflict with the question and existing reasoning steps. Furthermore, we discover that soft prompt tokens in certain positions have a strong impact on reasoning, and we can suppress the negative effects of the soft prompts on reasoning by corrupting them.

Inspired by the above findings, we propose a novel two-stage method called Dynamic Prompt Corruption (DPC): *Dynamic Trigger* and *Dynamic Corruption*. Concretely, in the first stage, we measure the impact factor of the soft prompts on the latter part of the reasoning in the final layer, by which we can differentiate whether the soft prompts can lead to the correct answer; in the second stage, we locate the key token that influences both the question and reasoning steps most in shallow layers, then mitigate the negative impact with key token vector masking and other token sparse masking.

To validate our analysis and evaluate the proposed DPC, we conduct comprehensive experiments with multiple LLMs on various tasks. Experimental results show that DPC can consistently improve the overall performance of LLMs such as LLaMA2-13B (Touvron et al., 2023), LLaMA3-8B (Dubey et al., 2024), and Mistral-0.2-7B (Jiang et al., 2023) on various reasoning tasks with 4%-8% accuracy enhancement including GSM8K (Cobbe et al., 2021), MATH (Hendrycks et al., 2021), AQuA (Ling et al., 2017) compared to pure prompt tuning. These results are in line with our observations and demonstrate the effectiveness of our DPC. Our contributions can be summarized as follows:

- We investigate the relationship between the information accumulation in the soft prompts and erroneous information flow patterns through saliency score analysis. Our findings indicate that when the soft prompts influence the later stages of reasoning deeply, the model is more likely to reach an incorrect answer.

- Based on the thorough analysis, we propose DPC, an instance-level prompt tuning optimization strategy, which can dynamically identify and mitigate the negative effects caused by the soft prompts.

- Experiment results manifest our proposed DPC can improve the vanilla prompt tuning for various LLMs on complex reasoning tasks to a large margin, demonstrating the effectiveness of our findings and the superiority of the proposed method.

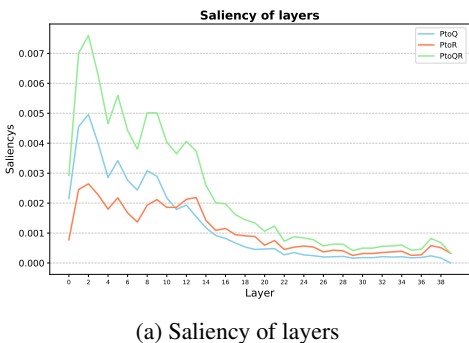

(a) Saliency of layers

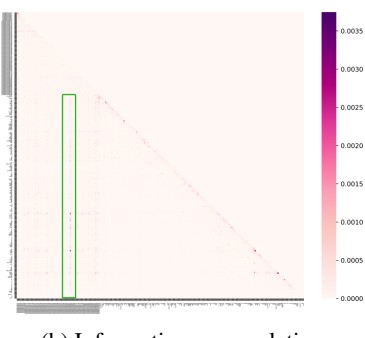

(b) Information accumulation

Figure 2: (a) Saliency scores of prompt-to-question, prompt-to-rationale layer by layer (b) Illustration of a significant information accumulation phenomenon within soft prompts, where a specific token conveys strong information to both the question and the rationale.

## 2 INFORMATION FLOW ANALYSIS FOR SOFT PROMPTS

To identify the influential factors in Chain-of-Thought (CoT) reasoning, we aim to analyze the model's inference process. A complete CoT reasoning instance consists of three primary components: the Prompt $\mathbf{P}$, the Question $\mathbf{Q}$, and the Rationale $\mathbf{R}$. Therefore, we need to examine the interactions among these three components using an appropriate tool. The saliency score, a widely used interpretative analysis tool employed to study the interactions between tokens, integrates gradient and attention values and is particularly effective in analyzing the information flow between different components. This allows the saliency score to simultaneously reflect the relationships among various modules and their impact on the output.

The saliency score is computed by taking the Hadamard product of the attention scores $\mathbf{A}$ and their gradients as follows:

$$\mathbf{S} = \left| \sum_h \left( \mathbf{A}_{h,l} \odot \frac{\partial \mathcal{L}(x)}{\partial \mathbf{A}_{h,l}} \right) \right| \tag{1}$$

where $\mathbf{A}_{h,l}$ denotes the attention weight for the $h$-th head in the $l$-th layer, $\odot$ represents the Hadamard product, and $\mathcal{L}(x)$ is the loss function, typically cross-entropy in our implementation. The attention matrix captures interactions within the sequence, while the gradient indicates the direction of information transfer. By analyzing the saliency score, we can thus observe the information flow throughout the entire sequence during reasoning.

### 2.1 LAYER ANALYSIS

Popular LLMs are composed of multiple stacked Transformer layers, which have specific roles in processing information during model reasoning. To better quantify the interactions between distinct components, we define two metrics to measure the strength of information flow layer-wise:

$\mathbf{S}_{pq}$, **the average significance of information flow from the prompt to question:**

$$\mathbf{S}_{pq}^{(\ell)} = \frac{\sum_{(i,j) \in \mathbb{C}_{pq}} \mathbf{S}_\ell(i,j)}{|\mathbb{C}_{pq}|} \tag{2}$$

$$\mathbb{C}_{pq} = \{(i,j) | p_s \leq i \leq p_e, q_s \leq j \leq q_e\} \tag{3}$$

$\mathbf{S}_{pr}$, **the average significance of information flow from the prompt to rationale:**

$$\mathbf{S}_{pr}^{(\ell)} = \frac{\sum_{(i,j) \in C_{pr}} \mathbf{S}_\ell(i,j)}{|\mathbb{C}_{pr}|} \tag{4}$$

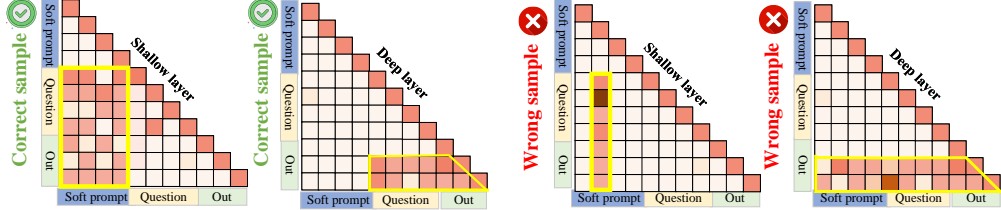

Figure 3: Schematic illustration of our observations. Correct-answer cases (left) show a balanced accumulation of saliency in shallow layers, where rationale information is evenly gathered from the soft prompts. As the reasoning progresses into deeper layers, attention shifts from the soft prompts to earlier rationale steps and the question itself. In contrast, wrong-answer cases (right) exhibit excessive saliency in shallow layers and disruptions in the information flow in deep layers. The latter part of the reasoning in deep layers overly focuses on the soft prompts, leading to incorrect answers.

$$\mathbb{C}_{pr} = \{(i,j)|p_s \leq i \leq p_e, r_s \leq j \leq r_e\} \tag{5}$$

where $\mathbf{S}_\ell(i,j)$ represents the intensity of information flow from the $i$-th to the $j$-th token, $|\mathbb{C}_{pq}|$ and $|\mathbb{C}_{pr}|$ denote the number of interactions between prompt and question tokens and between prompt and rationale tokens, respectively. Here, $p_s$, $q_s$, and $r_s$ are the start tokens of the prompt, question, and rationale, while $p_e$, $q_e$, and $r_e$ are their respective end tokens.

As shown in Figure 2a, we visualize the saliency scores during the LLM processing input across layers, sub-figures show the saliency scores representing the semantic information transfer between different input components: from the prompt to the question, and from the prompt to the rationale. We find distinct peaks for $\mathbf{Q}$ and $\mathbf{R}$ related to the soft prompts $\mathbf{P}$ occurring between layer 2 and layer 10, indicating that the most intense flow of prompt information is aggregated in shallow layers (near the input). Furthermore, we need a fine-grained analysis of the information flow phenomenon in shallow layers to explore the underlying mechanisms more precisely.

Figure 2b illustrates the saliency matrix $\mathbf{S}$ of shallow layers during reasoning, where we observe a strong flow of information among $\mathbf{P}$, $\mathbf{Q}$, and $\mathbf{R}$ in certain instances, leading to evident accumulation of information on a specific soft token. Such a phenomenon implies that certain specific soft tokens can have a significant impact on subsequent reasoning in some cases.

## 2.2 PROMPT TUNING ANALYSIS

After examining numerous instances, we find that in wrong-answer cases, the accumulation of saliency in shallow layers is often accompanied by changes in the information flow patterns in deep layers, as shown in Figure 3. In contrast, in correct-answer cases, reasoning gathers rationale information evenly from soft prompts in shallow layers, while in deeper layers, the latter part of the reasoning steps focuses on earlier rationale steps and the question rather than soft tokens. In incorrect examples, information accumulation occurs in shallow layers, accompanied by changes in deep information flow patterns: the latter part of the reasoning in deep layers overly focuses on the soft prompts, leading to incorrect answers.

### 2.2.1 WHAT IS THE RELATIONSHIP BETWEEN INFORMATION ACCUMULATION AND CHANGES IN INFORMATION FLOW PATTERNS?

We observe a potential correlation between information accumulation and changes in information flow patterns through visualization; however, the phenomenon in a few cases may not confirm a universal pattern. In this part, we continue to explore the impact of information accumulation on the information flow pattern.

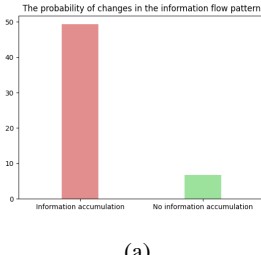 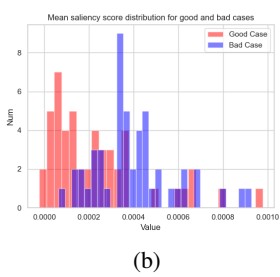 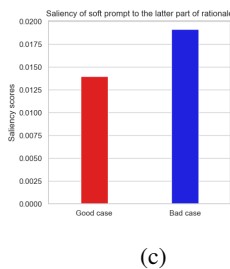

(a)             (b)             (c)

Figure 4: (a) describes the relationship between information accumulation in shallow layers and the change of information flow patterns in deep layers. (b) illustrates the distribution of information flow intensity from soft prompts to the latter part of the rationale in both good and bad cases. In good cases, the distribution tends to be weaker, while in bad cases, the distribution is stronger. (c) depicts the overall intensity of information flow from the soft prompts to the latter part of the rationale for both good and bad cases.

**Setting** The information accumulation can be interpreted as a specific token strongly influencing the subsequent reasoning process, and it is visible in the saliency matrix, where a distinct column effect can be observed. To explore the information accumulation, we compute the saliency scores with soft prompt token $i$ to question and rationale:

$$S_{p_iqr}^{(\ell)} = \sum_{(j,i)\in C_{p_iqr}} S_\ell(j,i) \tag{6}$$

$$\mathbb{C}_{p_iqr} = \{(j,i)|q_s \leq j \leq r_e\} \tag{7}$$

where $S_\ell(j,i)$ denotes the intensity of information flow from the $j$-th to the $i$-th token, $q_s$ is the start token of the question and $r_e$ is end token of the rationale. When $I_{acc} = \{p_s \leq i \leq p_e \mid S_{p_iqr}^{(l)} > \alpha \cdot \sum_{i=p_s}^{i=p_e} S_{p_iqr}^{(l)}/(p_e - p_s)\}$, where $\alpha = 10$ by default, is present, we consider that the information accumulation has occurred. When the information flow intensity between the latter reasoning steps and the soft prompts becomes significantly strong, we consider this a change in the information flow pattern. To confirm the existence of the information flow change, we calculate $S_{IFP}$:

$$S_{IFP} = \frac{\sum_{(i,j)\in C_{pr}} S_{last}(i,j)}{|C_{pr}|} \tag{8}$$

$$C_{pr} = \{(i,j)|p_s \leq i \leq p_e, r_h \leq j \leq r_e\} \tag{9}$$

where $S_{last}(i,j)$ represents the intensity of information flow from the $i$-th to the $j$-th token In the last layer of the model, $|C_{pr}|$ denotes the number of interactions among prompt tokens and rationale tokens, $p_s$ is the start token of the soft prompts, while $r_h$ corresponds to the intermediate token. Similarly, $p_e$ and $r_e$ denote the end tokens. If $S_{IFP}$ exceeds the defined threshold, we consider it a change in the information flow pattern.

**Observation** We randomly sample one hundred examples to validate the generality of these patterns on a larger scale. As shown in Figure 4a, the probability of information accumulation in the shallow layers occurring alongside changes in the deep layer information flow patterns is much higher than in cases where no accumulation is observed, indicating a strong correlation between the occurrence of information accumulation and changes in the deep layer information flow patterns.

### 2.2.2 WHAT DO CHANGES IN INFORMATION FLOW PATTERNS AFFECT?

The observation of the previous subsection proves the relation between information accumulation and information flow pattern changes. However, the impact of these changes on the model's reason-

ing ability remains unclear. Therefore, in this subsection, we delve into the influence of the changes on the accuracy of reasoning.

**Setting** We randomly selected 50 correct-answer and 50 incorrect-answer instances to demonstrate the applicability of the changes across a broader range. By calculating the $S_{IFP}$, we aim to measure the relation between reasoning accuracy and the changes.

**Observation** Figure 4b elaborates the distribution differences in saliency values between correct and incorrect examples. As shown in Figure 4c, the latter part of the reasoning process exhibits a lower mean value for soft prompts in correct-answer examples compared to incorrect-answer ones, which supports the broader context of the above phenomenon. Thus, we can conclude that: ***soft prompts lead to reaching the right answer when the latter reasoning steps put more emphasis on former tokens compared to soft tokens. In contrast, when the soft prompts significantly impact the latter reasoning steps, the model may reach an incorrect answer***. These findings are in line with the human cognitive process: before answering a question, humans always recall the background knowledge related to the question and obtain some relevant inspiration to answer the question; As a carrier containing task-related knowledge (Xiao et al., 2024) (Yu et al., 2024), the soft prompts can provide beneficial hints to help models understand the question and generate the initial reasoning steps, then use the question and generated steps to solve the problem.

## 3 METHOD

In this section, we retrospect the vanilla Prompt Tuning and propose the Dynamic Prompt Corruption (DPC) method derived from our analysis and findings to enhance reasoning capabilities.

### 3.1 PROMPT TUNING

Prompt tuning (PT) is a paradigm where a pre-trained language model $\Theta$ can be fine-tuned towards a task $\mathcal{T}$ with training data $(X, Y) = \{x_i, y_i\}_{i=1}^N$, requiring only a few learnable vectors (soft prompts) that are prepended to the input embeddings while keeping the model parameters $\Theta$ frozen (Lester et al., 2021a). Specifically, for an input sequence $T = \{t_1, t_2, \ldots, t_n\} \in \mathbb{R}^{n \times d}$, PT adds a learnable prompt matrix $\mathbb{V} = \{v_1, v_2, \ldots, v_l\} \in \mathbb{R}^{l \times d}$, where $l$ is a hyperparameter. The loss function for PT is optimized to $P$ as follows:

$$\mathcal{L}_{PLM} = -\sum_i \log P(y_i | x_i; \Theta, P).$$
(10)

where the input can be represented by the concatenated matrix $[\mathbb{V}; T] \in \mathbb{R}^{(l+n) \times d}$.

### 3.2 INTERFERING FACTORS RECOGNITION

By analyzing the information flow during reasoning, we believe that the latter reasoning steps should focus on the reasoning tokens that have been generated rather than soft tokens. Therefore, it is essential to identify those reasoning processes in which the flow of erroneous information occurs. We think that incorrect reasoning is characterized by some affected tokens in the latter half of the reasoning sequence, and affected tokens are those that have high information flow with the soft prompts. If the proportion of affected tokens exceeds a certain ratio, it is considered likely to result in bad reasoning. Consequently, we calculate the proportion **R** of affected tokens within reasoning as follows:

$$R = \frac{\sum_{r_h}^{r_e} 1(S_{pr(i)} > \beta)}{\sum_{r_h}^{r_e} 1}$$
(11)

$$S_{pr(i)} = \frac{\sum_{(i,j) \in C_{pr(i)}} S_{last}(i,j)}{|C_{pr(i)}|}$$
(12)

$$C_{pr(i)} = \{(i,j) | p_s \le j \le p_e\}.$$
(13)

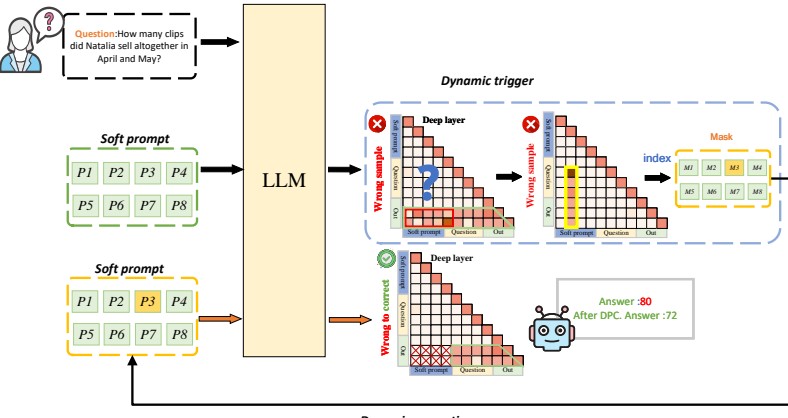

Figure 5: Overview of our method, Dynamic Prompt Corruption (DPC), which dynamically identifies erroneous information flow patterns and corrupts soft prompt tokens based on the location of information accumulation, alleviating the negative effects caused by the soft prompts in certain situations.

where $\beta$ is the average intensity of information flow from the soft prompts to the former part of the reasoning tokens by default, $S_{pr(i)}$ is the intensity of information flow from the soft prompts to the $i$-th token, $S_{last}(i, j)$ represents the intensity of information flow from the $i$-th to the $j$-th token in the model's final layer, $|C_{pr(i)}|$ is the number of interactions among prompts and token $i$, $p_s$ denotes the start token of the soft prompts, while $r_h$ corresponds to the intermediate token. Similarly, $p_e$ and $r_e$ denote the end tokens. After engaging in a comparative analysis of numerous instances from various datasets with distinct LLMs, we summarize different thresholds to delimit incorrect reasoning.

### 3.3 DYNAMIC CORRUPTION

With the aforementioned dynamic trigger strategy, we can identify the reasoning that produced incorrect answers, and we also need to push them to be correct. Inspired by the analysis of section 2.2.1, we propose the Dynamic Corruption strategy to pinpoint the soft prompt locations that cause erroneous information flow at the instance level and then mitigate these errors through targeted corruption operations. Firstly, we locate the position $j$ within the soft prompts where information accumulation is most intense by employing the calculation outlined in Equation 6. Secondly, we mask the value of the $j$-th prompt vector to obtain the corrupted soft prompts $t_c = \{v_1, v_2, \text{mask} \times v_j, \ldots, v_n\}$, and eliminate the smallest $\Gamma$ percent of the embedding values, excluding the $j$-th element. By default, $\Gamma$ is set to 10. Through the two steps, we can mitigate the potential negative effects of the harmful accumulation.

Combining the thorough analysis in the last section, we propose a novel two-stage strategy called Dynamic Prompt Corruption (DPC): Stage 1, identifying the negative impacts of soft tokens on the reasoning at the instance level; Stage 2, dynamically applying Corruption to the soft prompts, ultimately improving reasoning performance.

## 4 EXPERIMENT

### 4.1 SETTINGS

**Models** We test DPC and comparison methods on LLaMA3-8B (Dubey et al., 2024), LLaMA2-13B (Touvron et al., 2023), and Mistral-0.2-7B (Jiang et al., 2023) since they are popular Transformer decoder-only LLMs that are convenient for exploiting and analyzing inside architectures. We set the generation mode to greedy decoding to minimize irrelevant confounders during model inference and to ensure consistent answers to fixed questions under the same model and prompt; all experiments are run on an NVIDIA A100 GPU server.

**Datasets** GSM8K (Cobbe et al., 2021) is a challenging dataset for assessing the capability of language models in multi-step math reasoning. MATH (Hendrycks et al., 2021) is a comprehensive collection of high school-level mathematics problems. It covers a wide range of topics, including algebra, geometry, calculus, and more advanced subjects like number theory and combinatorics. AQuA (Ling et al., 2017) is designed for algebraic question-answering tasks and includes a variety of multiple-choice math word problems. The questions in AQuA are derived from standardized tests and other educational resources, focusing on algebra and related topics.

**Baselines** To validate the effectiveness of the proposed method, we conduct experiments with several baseline models that exhibit superior performance in general NLP tasks.

*Prompt tuning*: Tune the trainable prompt with the pretrained model frozen. As the model is fixed, prompt tuning will not change the pretrained model's performance on in-context learning tasks compared to when the prompt is removed.

*ACT*: Attention Calibration Technique(ACT) (Yu et al., 2024) is a training-free approach aimed at optimizing the attention distributions within LLMs without requiring weight fine-tuning. It is designed to address the phenomenon of attention sinks that receive disproportionately high attention despite their limited semantic relevance. By visualizing and analyzing the attention patterns across LLMs during inference, ACT identifies and calibrates attention sinks in an input-adaptive manner.

## 4.2 RESULTS AND DISCUSIONS

Table 2 shows the reasoning results with LLaMA3-8B, Llama2-13B, and Mistral-0.2-7B using various methods on 3 complex reasoning tasks.

**GSM8K** Compared with vanilla prompt tuning, our Dynamic Prompt Correction (DPC) method enhanced the accuracy on the GSM8K dataset for Llama2, Llama3, and Mistral-0.2, improving from 38.1%, 65.5%, and 49.5% to 41.9%, 67.6%, and 51.1%, respectively. Notably, vanilla prompt tuning performed suboptimally with Llama3, showing a mere 0.6% increase in accuracy over the pretrained model. In contrast, our DPC method mitigated the challenges of vanilla prompt tuning, achieving a 2.7% improvement in accuracy. Notably, although the ACT method has shown impressive performance in classification tasks, its generalization capability in complex multi-step reasoning tasks is limited, and it fails to alleviate the challenges associated with prompt tuning.

**MATH** For the MATH dataset, which contains challenging competition-level mathematics problems, models typically perform poorly due to the task's difficulty. However, our DPC method improved the accuracy of Llama2, Llama3, and Mistral-0.2 on the MATH dataset, increasing from 7.6%, 33.7%, and 15.0% to 9.2%, 36.3%, and 16.4%, respectively. This demonstrates the superior performance of our method on difficult tasks.

**AQuA** DPC improves the accuracy on the AQuA dataset for Llama2, Llama3, and Mistral-0.2 from 22.4%, 33.7%, and 28.7% to 31.1%, 42.5%, and 31.9%, respectively. Notably, we observe an average improvement of 6.9%, and a maximum improvement of 8.8%, showing its outstanding performance in multi-step reasoning multiple-choice tasks.

Table 1: Accuracy % of various baselines and DPC on 3 tasks with LLaMA3-8B, Llama2-13B and Mistral-0.2-7B.

| | Llama2-13B | | | Llama3-8B | | | Mistral-7B | | |
|---|---|---|---|---|---|---|---|---|---|
| | GSM8K | MATH | AQuA | GSM8K | MATH | AQuA | GSM8K | MATH | AQuA |
| Pretrained model | 29.5 | 2.0 | 21.0 | 64.9 | 30.0 | 34.0 | 37.9 | 5.1 | 26.0 |
| Prompt tuning | 38.1 | 7.6 | 22.4 | 65.5 | 33.7 | 38.5 | 49.5 | 15.0 | 28.7 |
| ACT | 39.2 | 7.1 | 20.1 | 52.6 | 33.8 | 38.6 | 49.5 | 15.0 | 28.7 |
| DPC | **41.9** | **9.2** | **31.1** | **67.6** | **36.3** | **42.5** | **51.1** | **16.4** | **31.9** |

## 4.3 ABLATION STUDIES

To evaluate the effectiveness of various components in our proposed Dynamic Prompt Correction (DPC) method, we conducted a series of ablation experiments. Specifically, we investigated the impact of corruption without Dynamic Trigger and random corruption. By systematically removing or altering some elements, we aim to analyze their contributions to the overall performance and identify key factors that lead to improved reasoning and reduced error rates. The following experiments explore the role of Dynamic corruption and Dynamic Trigger in addressing information accumulation and erroneous information flow patterns.

**Setting 1:All corruption**    We applied the corruption operation to all instances without the need for confirmation from the dynamic trigger. This allows us to evaluate the effectiveness of the dynamic trigger in isolation, providing insights into its overall impact on model performance.

**Setting 2:Random corruption**    For each instance of the corruption operation, we randomly selected the position to be corrupted. This random selection enables us to assess the overall robustness of dynamic corruption, and provides a more comprehensive understanding of its effects on model performance.

Table 2: Accuracy % of various baselines and DPC on 3 tasks with LLaMA3-8B.

| Model | LLaMA3-8B | | |
| --- | --- | --- | --- |
| | GSM8K | MATH | AQuA |
| Prompt tuning | 65.5 | 33.7 | 38.5 |
| Random corruption | 51.3 | 32.3 | 39.3 |
| All corrution | 65.8 | 34.2 | 39.9 |
| DPC | **67.6** | **36.3** | **42.5** |

The results of the ablation studies, presented in Table 2, provide crucial insights into the combined effectiveness of dynamic trigger and dynamic corruption in our Dynamic Prompt Correction (DPC) method. The performance of DPC is compared against several baselines, including prompt tuning, random corruption, and corruption without dynamic trigger.

**Effectiveness of Random Corruption:**    Random corruption, where the position of the corruption is selected randomly for each instance, shows a drop in accuracy across all tasks compared to prompt tuning. Specifically, the accuracy on GSM8K, MATH, and AQuA decreases to 51.3%, 32.3%, and 39.3%, respectively. This demonstrates that while corruption can introduce changes, applying it without specific guidance (such as from a dynamic trigger) can disrupt critical information flow, leading to worse performance.

**Effectiveness of Corruption Without Dynamic Trigger:**    Applying corruption to all instances without the dynamic trigger results in marginal improvements over prompt tuning, particularly on the MATH and AQuA datasets, with accuracy increasing to 34.2% and 39.9%, respectively. However, the improvement is modest, and the GSM8K task shows only a minor gain (65.8% vs. 65.5%). These results suggest that although corruption can mitigate some negative effects, the absence of a dynamic trigger to guide when and where to apply it limits its full potential.

**Dynamic Prompt Corruption - Combined Effect of Dynamic Trigger and Dynamic Corruption:**    The full DPC method, which integrates both dynamic trigger and dynamic corruption, demonstrates significant performance gains across all tasks. For GSM8K, MATH, and AQuA, DPC achieves 67.6%, 36.3%, and 42.5% accuracy, respectively. These results highlight that the dynamic trigger plays a vital role in identifying when and where to apply corruption effectively, while dynamic corruption corrects erroneous information flows precisely. The combination of these two components is essential to achieving optimal performance, as neither component alone is sufficient to address the complexities of the task.

**Conclusion:**    The ablation study clearly shows that dynamic corruption alone, or applied indiscriminately, is not sufficient to significantly improve model performance. It is the interaction be-

tween dynamic trigger and dynamic corruption that yields the best results. The dynamic trigger ensures that corruptions are applied only when necessary, while dynamic corruption addresses specific errors, leading to a more robust reasoning process and higher overall accuracy. The synergy between these two components is key to the success of the DPC method.

## 5 RELATED WORK

Parameter-efficient fine-tuning (PEFT) (Han et al., 2024; Ding et al., 2023b) aims to tune a few parameters while matching the performance of full model fine-tuning. PEFT methods fall into two main branches. The first approximates parameter changes in lower-dimensional spaces. For instance, LoRA (Hu et al., 2022) decomposes $W \approx W_0 + AB$, where $W_0 \in \mathbb{R}^{d_1 \times d_2}$ is the original weight matrix, and $A \in \mathbb{R}^{d_1 \times r}, B \in \mathbb{R}^{r \times d_2}$ are low-rank matrices capturing fine-tuning updates. Other examples include Bitfit (Ben Zaken et al., 2022), tuning only bias terms of PLMs, Diff Pruning (Guo et al., 2021), learning task-specific "diff" vectors to extend pre-trained parameters, and Adapter (Houlsby et al., 2019), an early method adding task-specific layers. The second branch leverages Transformer (Vaswani et al., 2017) structure, treating input tokens or prefixes as trainable parameters. Prompt Tuning (PT) (Lester et al., 2021b) prepends learnable prompt tokens to inputs, while Prefix-tuning (Li & Liang, 2021) tunes intermediate hidden states. P-tuning (Liu et al., 2023) and P-tuning v2 (Liu et al., 2022) extend these ideas with varied designs.

As a key PEFT method in the second branch, Prompt Tuning (PT) (Lester et al., 2021b) is valued for its lightweight and simple implementation, widely applied in vision and language tasks. Subsequent studies have refined it: (Oymak et al., 2023) analyzed attention's role, (Yuan et al., 2024) adapted prompts per query, and (Petrov et al., 2024) found soft prompting biases attention heads, limiting it to pre-trained skills. (Wang et al., 2023b) showed PT acts as a universal approximator for sequence-to-sequence tasks, though its expressive power is capped, unlike MLP. Additional research includes subspace tuning (Qin et al., 2021; 2024), two-stage pipelines (Wang et al., 2024), Low-norm Effect analysis (Fu et al., 2024) showing norm reduction boosts performance, prompt decomposition (Shi & Lipani, 2024) for efficiency—our focus here—and a dynamic PT framework (Yang et al., 2023).

## 6 CONCLUSION

In this work, we explored the limitations of vanilla Prompt Tuning in complex reasoning tasks and uncovered key factors that contribute to its underperformance. Through neuron saliency score analysis, we demonstrated how soft prompts could positively or negatively affect reasoning processes, depending on how they influence subsequent reasoning steps. Inspired by these insights, we introduced Dynamic Prompt Corruption, a novel approach designed to mitigate the negative impacts of soft prompts on complex reasoning tasks. By employing a two-step process involving dynamic trigger and dynamic corruption, our method successfully adjusts the influence of soft prompts in a way that guides the model towards correct reasoning. Comprehensive experiments across various models and datasets validated the effectiveness of our approach. These findings and results contribute to a deeper understanding of soft prompting mechanisms and provide a practical solution for enhancing reasoning tasks in large language models.

## LIMITATIONS

In this work, DPC has demonstrated strong performance; however, the process of identifying erroneous information flow patterns and pinpointing the critical soft prompt tokens requires a complete reasoning pass, which increases the inference cost in practical applications. While our research provides valuable insights into the underlying mechanisms of soft prompts, it should not be viewed as the sole interpretation. We believe that more effective methods may exist to further explain these phenomena.

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

# A IMPLEMENT DETAILS

## A.1 PROMPT TUNING TRAINING SETUP

For all prompt tuning experiments, we use a learning rate of 0.001 and a prompt length of 100. For the LLama3-8b model, we initialized the soft prompts randomly. For LLama2-13b and Mistral-0.2-7b, we initialized the soft prompts with text. The text used for initialization was selected from a single example in the training set of each dataset: GSM8K, MATH, and AQuA. For each dataset, one representative sample, including both the problem and its rationale, was chosen to initialize the soft prompts. This ensures that the prompts are aligned with the specific characteristics of each dataset, enabling more effective prompt tuning during training.

For GSM8K :

```
Question:  Natalia sold clips to 48 of her friends in April,
and then she sold half as many clips in May.  How many clips did
Natalia sell altogether in April and May?
```

Answer:  Natalia sold $\frac{48}{2} = 24$ clips in May.  Natalia sold $48 + 24 = 72$ clips altogether in April and May.  #### 72

For MATH :

```
Problem:
```
What is the degree of the polynomial $(4 + 5x^3 + 100 + 2\pi x^4 + \sqrt{10}x^4 + 9)$?

```
Solution:
```
This polynomial is not written in standard form. However, we don't need to write it in standard form, nor do we need to pay attention to the coefficients. We just look for the exponents on $x$. We have an $x^4$ term and no other term of higher degree, so $\boxed{4}$ is the degree of the polynomial.

For AQuA:

```
Question:  A person is traveling at 20 km/hr and reached his
destination in 2.5 hr.  Find the distance?

A) 53 km
B) 55 km
C) 52 km
D) 60 km
E) 50 km
```
$T = 2.5$ hrs $= \frac{5}{2}$ hrs
$D = T * S = 20 \times \frac{5}{2} = 50$ km

```
Answer is E
```

# B  COMPARISON OF METHODS

Firstly, as shown in Figure 6(a), our analysis tool differs from that used in StreamingLLM(Xiao et al., 2024). While StreamLLM uses attention scores, we utilize saliency scores, which combine gradients and attention values. This allows us to effectively analyze information flow between different components, making it suitable for scenarios like ours that involve multiple components. This methodological difference may lead to different analysis results.

Additionally, as illustrated in Figure 6(b), StreamingLLM observes high attention values on the first token, termed an "attention sink," and leverages this finding to extend the input sequence length—a positive use of high attention values.

However, different scenarios can exhibit different behaviors. For example, as shown in Appendices B Figure 6(c), (d), and (e), several studies have demonstrated the negative effects of the attention sink phenomenon.

The ACT(Yu et al., 2024) study found that attention sinks not only occur on the first token but also on tokens with limited semantic information (e.g., ".", ":", and "<0x0A>"), as visualized in Figure 6(c). Contrary to the observations made in StreamingLLM—which suggest preserving attention sinks to enhance LLMs' accuracy—they highlight that not all attention sinks are beneficial. Specifically, for most attention sinks occurring in the middle or later parts of inputs, reducing their attention scores can lead to improved accuracy.

In OPERA(Huang et al., 2023) and DOPRA(Wei & Zhang, 2024), as represented in Figure 6(d)(e), studies found that when models generate hallucinated content, the self-attention weights in the last layer exhibit a distinct "columnar" pattern before the hallucination occurs, leading to an "over-trust" tendency in the self-attention weights for the hallucinated parts.

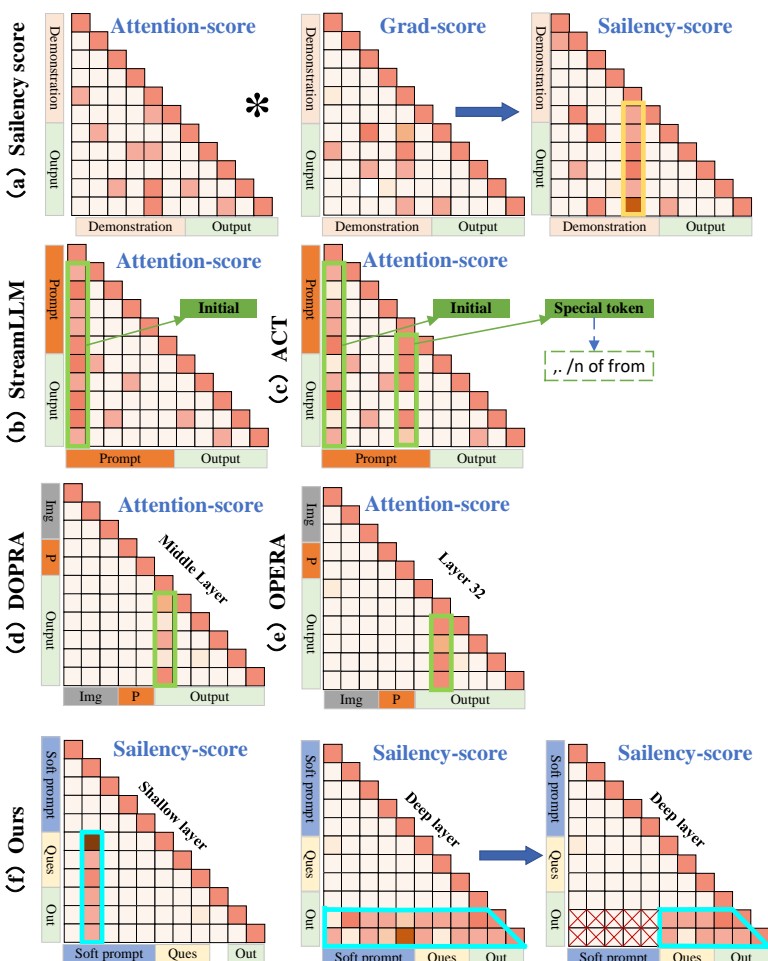

Figure 6: Schematic illustration of methods related to information accumulation: (a) illustrates the distinction between saliency and attention; (b) demonstrates the attention sink phenomenon on initial tokens as described in StreamingLLM; (c) highlights the attention sink phenomenon observed in ACT for special tokens other than initial tokens; (d) and (e) present the accumulation phenomenon related to hallucinations discovered by DOPERA and OPERA in the visual domain; (f) showcases the information accumulation phenomenon we identified in soft prompts.

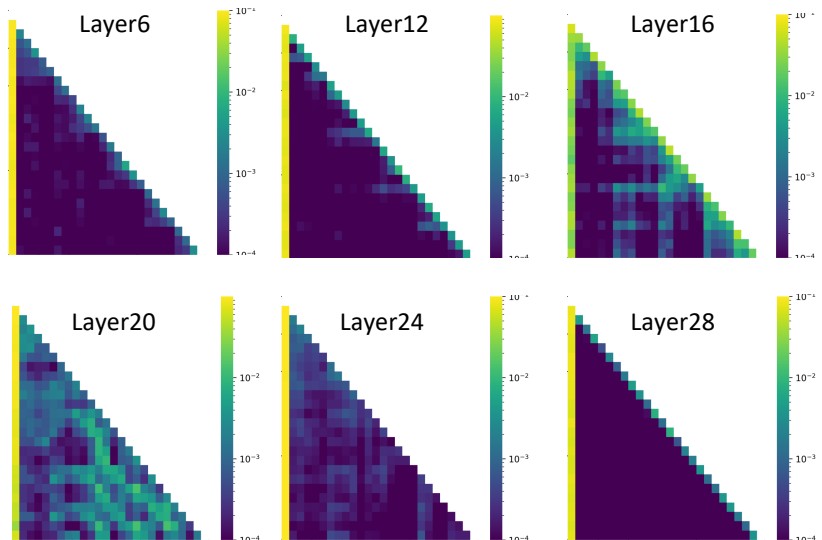

Figure 7: Visualization of the attention matrix in Llama3-8b.

## C    VISUALIZATION OF THE ATTENTION MATRIX

## D    COMPLEMENTARY EXPERIMENTS

To further demonstrate the effectiveness of our method, we have added additional comparisons with other PEFT methods. We conducted extensive experiments using prefix tuning and LoRA, and our DPC method consistently showed improvements within the scope of prompt tuning. Furthermore, Table 3 indicates that DPC achieves optimal or near-optimal performance with relatively few computational resources compared to multiple methods, confirming its effectiveness.

Table 3: Accuracy % of various peft methods and DPC on 3 tasks with LLaMA3-8B, Llama2-13B and Mistral-0.2-7B.

|  | Llama2-13B | | | Llama3-8B | | | Mistral-7B | | |
| --- | --- | --- | --- | --- | --- | --- | --- | --- | --- |
|  | GSM8K | MATH | AQuA | GSM8K | MATH | AQuA | GSM8K | MATH | AQuA |
| Pretrained model | 29.5 | 2.0 | 21.0 | 64.9 | 30.0 | 34.0 | 37.9 | 5.1 | 26.0 |
| Prompt tuning | 38.1 | 7.6 | 22.4 | 65.5 | 33.7 | 38.5 | 49.5 | 15.0 | 28.7 |
| Prefix tuning | 41.7 | 8.4 | 20.1 | 65.4 | 33.0 | 41.3 | **54.4** | 16.3 | 31.5 |
| LoRA | 12.7 | 7.4 | 24.8 | 44.1 | 31.8 | 38.6 | 44.9 | **19.9** | 24.4 |
| DPC | **41.9** | **9.2** | **31.1** | **67.6** | **36.3** | 42.5 | 51.1 | 16.4 | **31.9** |

To further verify the generalizability of our method, we conducted experiments on another three datasets involving logical reasoning and commonsense reasoning. Table 4 demonstrates that DPC not only excels in the initial tasks but also effectively generalizes to other reasoning scenarios.

Table 4: Accuracy % of prompt tuning and DPC on another 3 tasks with LLaMA3-8B and Mistral-7B.

|  | Llama3-8B | | | Mistral-7B | | |
| --- | --- | --- | --- | --- | --- | --- |
|  | prontoQA | ARC-challenge | WorldTree | prontoQA | ARC-challenge | WorldTree |
| Prompt tuning | 100 | 13.2 | 39.1 | 97.9 | 14 | 55.2 |
| DPC | **100** | **14.8** | **45.5** | **98.9** | **16.6** | **56.5** |

