# OpenReview forum: "Improving Complex Reasoning with Dynamic Prompt Corruption: A Soft Prompt Optimization Approach"
_ICLR.cc/2025/Conference — ICLR 2025 Poster_

### Official Review · Reviewer_fh4M · 2024-10-27

**Soundness:** 3
**Presentation:** 4
**Contribution:** 2
**Rating:** 6
**Confidence:** 4

**Summary:**

This paper discovers that sometimes prompts can make the model generate the wrong answer, using the metric of saliency scores defined by Hadamard product of attention score and its gradients. When such cases happen, it is usually because the question part does not have even attention on the prompt (i.e., not reading hints carefully), or because the answer part relies too much on the prompt (i.e., misled by the hints, rather than thinking more on questions and previous steps in answer). Therefore, the authors proposed that one can use two step algorithm to fix. In the first step, they identity whether such bad cases happen. In the second step, they fix bad cases by adding masks to prompts, so that attention will be improved. Empirically such method can improve the performance on GSM8K, MAH, and AQuA.

**Strengths:**

Originality: the authors noticed that the attentions between soft prompts and question, and between soft prompts and answer, are nicely correlated with the final correctness. This observation is interesting and original.

Quality: the observation is interesting. I especially like the visualizations in the paper.

Clarity: this paper is well written and easy to follow.

Significance: the observation that attentions will affect performance might be interesting to other researchers.

**Weaknesses:**

I will not give a high score to the quality of this paper, because it did not tell a "full story".
- The Figure 4 indeed shows difference between good and bad cases of the saliency scores, but the difference is not very big. For example, in Figure 4(c), good case has score of ~0.014, while bad case has score of ~0.019. While I agree such difference means we can improve our model by using the two step algorithm, this observation does not seem to be substantial. It seems entirely possible that in the next version of LLM, this difference may disappear.
- While I agree the two step algorithm is effective to improve the model's accuracy, the improvement is not significant as well.

Therefore, I think the significance of this paper is mild. I feel this algorithm may not be used by most researchers in the field.

**Questions:**

I do not have additional questions.

---

> ### Author Response · Authors · 2024-11-21
> **Response to Reviewer fh4M**
>
> Thank you for the constructive advice and comments, and we are glad to reply to your invaluable suggestions and questions.
>
> ### **Response to Weaknesses**
>
> **Weakness 1:**
> The Figure 4 indeed shows difference between good and bad cases of the saliency scores, but the difference is not very big. For example, in Figure 4(c), good case has score of ~0.014, while bad case has score of ~0.019. While I agree such difference means we can improve our model by using the two step algorithm, this observation does not seem to be substantial. It seems entirely possible that in the next version of LLM, this difference may disappear.
>
> **Response to Weakness 1:**
>
> Regarding your comment, "Figure 4 indeed shows a difference between good and bad cases of the saliency scores, but the difference is not very big. For example, in Figure 4(c), the good case has a score of ~0.014, while the bad case has a score of ~0.019," we need to emphasize that Figure 4(c) shows the difference between the mean values of the two distributions in Figure 4(b). Although the difference in Figure 4(c) may seem small in absolute terms, the statistical difference between the two distributions in Figure 4(b) is highly significant. The distributions differ not only in their means but also in their overall shape and spread.
>
> Regarding your concern that the observed phenomenon may disappear in the next version of LLMs, we have conducted experiments and found that this phenomenon still exists from Llama2 to Llama3. While we agree that next-generation models are becoming more powerful, their exact performance characteristics remain unknown at this time. However, we want to point out that our method does not require modifying the internal structure of the model. This design ensures the reusability and adaptability of our approach, making it applicable even with more advanced language models in the future. Therefore, even as models evolve, our approach remains relevant and can be readily applied to new architectures without significant modifications.
>
>
>
> **Weakness 2:**
> While I agree the two step algorithm is effective to improve the model's accuracy, the improvement is not significant as well. Therefore, I think the significance of this paper is mild.
>
> **Response to Weakness 2:**
>
> To further demonstrate the effectiveness of our method, we have added additional comparisons with other PEFT methods. We conducted extensive experiments using prefix tuning and LoRA, and our DPC method consistently showed improvements within the scope of prompt tuning. Furthermore, our results indicate that DPC achieves optimal or near-optimal performance with relatively few computational resources compared to multiple methods, confirming its effectiveness.
>
> Beyond performance improvements, our work also offers novel insights to the fields of prompt tuning and PEFT. We hope to inspire more researchers to focus on the information flow trends among different input components and to uncover possible inherent connections.
>
>
>
> |                      |           | Llama2   |          |           | Llama3   |          |           | Mistral  |          |
> | -------------------- | --------- | -------- | -------- | --------- | -------- | -------- | --------- | -------- | -------- |
> | **Method/Datasets**  | **GSM8k** | **MATH** | **AQuA** | **GSM8k** | **MATH** | **AQuA** | **GSM8k** | **MATH** | **AQuA** |
> | **Pretrained model** | 29.5      | 2.0      | 21.0     | 64.9      | 30.0     | 34.0     | 37.9      | 5.1      | 26.0     |
> | **Prompt tuning**    | 38.1      | 7.6      | 22.4     | 65.5      | 33.7     | 38.5     | 49.5      | 15.0     | 28.7     |
> | **Prefix tuning**    | 41.7      | 8.4      | 20.1     | 65.4      | 33.0     | 41.3     | **54.4**  | 16.3     | 31.5     |
> | **LoRA**             | 12.7      | 7.4      | 24.8     | 44.1      | 31.8     | 38.6     | 44.9      | **19.9** | 24.4     |
> | **DPC**              | **41.9**  | **9.2**  | **31.1** | **67.6**  | **36.3** | **42.5**   | *51.1*    | *16.4*   | **31.9** |
>
> The **bolded numbers** are the best results, and the *italic numbers*  represent the second-best results.

---

> > ### Comment · Reviewer_fh4M · 2024-11-29
> >
> > Thank you for the additional explanations and experiments. I will maintain my score unchanged.

---

> > > ### Author Response · Authors · 2024-12-01
> > >
> > > Thanks once again for your valuable comments.  We are very grateful for your consistently positive evaluation of our work.

---

> ### Author Response · Authors · 2024-11-24
> **Inquiry Regarding Rebuttal Feedback**
>
> Dear Reviewer fh4M,
>
> We sincerely appreciate your time and effort in reviewing our manuscript and offering valuable suggestions. We provided detailed clarifications in response to your questions a few days ago. If you have any additional feedback, concerns, or questions regarding our response, we would greatly appreciate hearing from you and welcome further discussion.

---

### Official Review · Reviewer_sfZi · 2024-10-29

**Soundness:** 3
**Presentation:** 3
**Contribution:** 3
**Rating:** 6
**Confidence:** 3

**Summary:**

This paper studies why sometimes soft prompts can affect reasoning outcomes. The paper points out that certain specific tokens in prompts can significantly harm reasoning performance. To address these challenges, it proposes a method called Dynamic Prompt Corruption (DPC), which seeks to optimize the use of soft prompts in reasoning tasks. DPC dynamically adjusts the influence of soft prompts based on their impact on the reasoning process. Specifically, it involves two key components: Dynamic Trigger and Dynamic Corruption. It mitigates the negative effects of detrimental soft prompts by selectively masking key tokens that interfere with the reasoning process.

**Strengths:**

1. The problem addressed in the paper is genuinely existing, and the paper has strong practical value.

2. The paper uses experiments to analyze in detail the reasons why prompts can lead to reasoning errors in certain situations, demonstrating strong persuasiveness and logical coherence.

3. The method and analysis proposed in the paper are highly relevant, simple and effective, and easy to reuse.

4. The paper is well-written and easy for readers to follow.

**Weaknesses:**

1. The choice of the saliency score lacks explanation; it's unclear why this metric is used instead of others. Additionally, there should be more clarification regarding Equation (1).

2. The phenomenon presented in the article is largely based on observation, but the generalizability of the observed conclusions requires further validation. Additionally, the explanations for the phenomenon lack theoretical support.

**Questions:**

Please see weaknesses.

---

> ### Author Response · Authors · 2024-11-21
> **Response to Reviewer sfZi**
>
> Thanks a lot for your valuable reviews, and we appreciate the time and effort you have taken. Regarding the weaknesses and questions, we would like to elaborate and address your concerns on this work.
>
> ### **Response to Weaknesses**
>
> **Weakness 1:**
>
> The choice of the saliency score lacks explanation; it's unclear why this metric is used instead of others. Additionally, there should be more clarification regarding Equation (1).
>
> **Response to Weakness 1:**
>
> There are several approaches for model interpretation, including attention-based methods, gradient-based methods, occlusion-based methods, and propagation-based methods. Each has its own advantages and is suitable for different scenarios.
>
> Saliency scores, which combine gradient and attention values, are particularly effective in analyzing the information flow between different components. Intuitively, the saliency score is only large when both the attention and gradient are large. This allows saliency to simultaneously reflect the relationships among various modules and their impact on the output, making it especially suitable for scenarios like ours, where multiple components are involved.
>
> We have added more clarification on this point in Equation (1) of the revised paper.
>
>
>
> **Weakness 2:**
>
> The phenomenon presented in the article is largely based on observation, but the generalizability of the observed conclusions requires further validation. Additionally, the explanations for the phenomenon lack theoretical support.
>
> **Response to Weakness 2:**
>
> Our "observation → hypothesis → validation" analysis method is inspired by some impactful works in the field, such as label words [1] and ACT [2]. This is a popular analytical paradigm in the domain. Meanwhile, as highlighted by Reviewer Y54T, we adopt a probing approach, quantitatively analyzing the efficacy of soft prompts in guiding complex reasoning within LLMs. We performed thorough qualitative and quantitative analyses to support our arguments, particularly through a step-by-step correlation analysis. The statistical correlation analysis can be considered as a theoretical justification.
>
> We begin with case-based observations of information flow within the model, as illustrated in Figure 3. Based on these findings, we propose a hypothesis: information accumulation in the shallow layers may correspond with  information flow pattern changes  in deep layers, and incorrect deep-layer flow patterns could lead to incorrect answers. This hypothesis is then substantiated through quantitative experiments. Figure 4(a) demonstrates a strong correlation between information accumulation and changes in deep-layer information flow patterns. Figures 4(b) and 4(c) further quantify the relationship between incorrect flow patterns and incorrect results by calculating  intensity of information flow  for the latter stages of reasoning in deep layers.
>
> Moreover, we employed a two-stage approach and conducted extensive experiments. The results showed consistent improvements, indicating that eliminating or reducing this phenomenon can enhance reasoning capabilities, thereby further validating our viewpoint.
>
> Ultimately, our findings confirm that soft prompts can guide models toward correct answers when later reasoning steps emphasize former tokens  more than soft tokens. Conversely, when soft prompts overly influence later reasoning steps, the model is more likely to produce incorrect answers. Beyond performance improvements, our work also offers novel insights to the fields of prompt tuning and PEFT. We hope to inspire more researchers to focus on the information flow trends among different input components and to uncover possible inherent connections.
>
>
>
> **Generalizability:**  To further verify the generalizability of our method, we conducted experiments on another three datasets involving logical reasoning and commonsense reasoning. Our results demonstrate that DPC not only excels in the initial tasks but also effectively generalizes to other reasoning scenarios.
>
> |                     |              | Llama3-8b         |               |              | Mistral-7b        |               |
> | ------------------- | ------------ | ----------------- | ------------- | ------------ | ----------------- | ------------- |
> | **Method/Datasets** | **prontoQA** | **ARC-challenge** | **WorldTree** | **prontoQA** | **ARC-challenge** | **WorldTree** |
> | **Prompt tuning**   | 100          | 13.2              | 39.1          | 97.9         | 14                | 55.2          |
> | **DPC**             | **100**      | **14.8**          | **45.5**      | **98.9**     | **16.6**          | **56.5**      |
>
>
>
> [1] Wang, Lean, et al. "Label Words are Anchors: An Information Flow Perspective for Understanding In-Context Learning."  emnlp2023.
>
> [2] Yu Z, Wang Z, Fu Y, et al. Unveiling and harnessing hidden attention sinks: Enhancing large language models without training through attention calibration. ICML24

---

> > ### Comment · Reviewer_sfZi · 2024-11-25
> >
> > These responses have effectively addressed my concerns, and I will maintain my positive evaluation of this paper.

---

> > > ### Author Response · Authors · 2024-11-25
> > >
> > > Thanks once again for your valuable comments. We are delighted to hear that your concerns have been addressed, and we are very grateful for your consistently positive evaluation of our work.

---

> ### Author Response · Authors · 2024-11-24
> **Inquiry Regarding Rebuttal Feedback**
>
> Dear Reviewer sfZi,
>
> We sincerely appreciate your time and effort in reviewing our manuscript and offering valuable suggestions. We provided detailed clarifications in response to your questions a few days ago. If you have any additional feedback, concerns, or questions regarding our response, we would greatly appreciate hearing from you and welcome further discussion.

---

### Official Review · Reviewer_Y54T · 2024-11-04

**Soundness:** 4
**Presentation:** 4
**Contribution:** 4
**Rating:** 8
**Confidence:** 4

**Summary:**

The paper introduces a novel technique called Dynamic Prompt Corruption (DPC) to optimize the utilization of soft prompts in intricate reasoning tasks. This method dynamically regulates the impact of soft prompts on the reasoning process. Experimental results demonstrate the effectiveness of this approach.

**Strengths:**

1. The athors adopt a probing approach, quantitatively analyzing the efficacy of soft prompts on guiding complex reasoning in LLMs. Its conclusions offer profound insights, and methods derived from these insights have demonstrated significant effectiveness in experiments. This work makes a substantial contribution to the field of LLM reasoning research, and I highly appreciate the innovative insights presented in this paper.

2. The article exhibits a clear and coherent structure, with well-organized content.

**Weaknesses:**

The experimental baselines in this study are relatively limited in number. A comparative analysis with other PEFT methods, like LORA, could enhance the experiments. If this paper can unveil shared observational conclusions among various PEFT methods, it would significantly strengthen the overall contribution of the study.

**Questions:**

1. Do other PEFT methods exhibit similar observations?
2. Apart from saliency scores, are there better probing tools available?

---

> ### Author Response · Authors · 2024-11-21
> **Response to Reviewer Y54T**
>
> We deeply appreciate the time and effort you invested in reviewing our paper, and we are glad to answer your questions.
>
> ### **Responses to Weaknesses**
>
> **Weakness :**
>
> The experimental baselines in this study are relatively limited in number. A comparative analysis with other PEFT methods, like LORA, could enhance the experiments. If this paper can unveil shared observational conclusions among various PEFT methods, it would significantly strengthen the overall contribution of the study.
>
> **Response to Weakness:**
>
> Following your suggestion, we conducted extensive PEFT experiments using prefix tuning, LoRA. Our DPC method consistently demonstrated improvements within the scope of prompt tuning. Furthermore, our results show that DPC achieves optimal or near-optimal performance with relatively few computational resources compared to multiple methods, confirming its effectiveness. Notably, we observed that LoRA sometimes led to performance degradation. We believe this may be because modern LLMs have undergone comprehensive post-training processes, including Supervised Fine-Tuning (SFT) and Reinforcement Learning from Human Feedback (RLHF). Consequently, fine-tuning model parameters on a single dataset can be challenging and may even result in decreased performance.
>
> Beyond performance improvements, our work also offers novel insights to the fields of  PEFT. We hope to inspire more researchers to focus on the information flow trends among different input components and to uncover possible inherent connections.
>
>
>
> |                      |           | Llama2   |          |           | Llama3   |          |           | Mistral  |          |
> | -------------------- | --------- | -------- | -------- | --------- | -------- | -------- | --------- | -------- | -------- |
> | **Method/Datasets**  | **GSM8k** | **MATH** | **AQuA** | **GSM8k** | **MATH** | **AQuA** | **GSM8k** | **MATH** | **AQuA** |
> | **Pretrained model** | 29.5      | 2.0      | 21.0     | 64.9      | 30.0     | 34.0     | 37.9      | 5.1      | 26.0     |
> | **Prompt tuning**    | 38.1      | 7.6      | 22.4     | 65.5      | 33.7     | 38.5     | 49.5      | 15.0     | 28.7     |
> | **Prefix tuning**    | 41.7      | 8.4      | 20.1     | 65.4      | 33.0     | 41.3     | **54.4**  | 16.3     | 31.5     |
> | **LoRA**             | 12.7      | 7.4      | 24.8     | 44.1      | 31.8     | 38.6     | 44.9      | **19.9** | 24.4     |
> | **DPC**              | **41.9**  | **9.2**  | **31.1** | **67.6**  | **36.3** | **42.5**   | *51.1*    | *16.4*   | **31.9** |
>
> The **bolded numbers** are the best results, and the *italic numbers*  represent the second-best results.
>
>
>
>
>
>
>
>
> ### **Responses to Question**
>
> **Question 1 :**
>
> Do other PEFT methods exhibit similar observations?
>
> **Response to Question 1:**
>
> Our analysis focuses on prompt tuning, which requires additional token length. Methods like LoRA might not be suitable for our scenario due to this requirement. However, our work can provide inspiration to LoRA by encouraging it to focus on the interactions between different parts of model parameters. Moreover, we believe that other PEFT methods that also introduce extra tokens, such as prefix tuning and P-tuning, can directly benefit from our work by focusing on the interactions between different input components. We will consider investigating this in future work.
>
> **Question 2 :**
>
> Apart from saliency scores, are there better probing tools available?
>
> **Response to Question 2:**
>
> There are several approaches for model interpretation, including attention-based methods, gradient-based methods, occlusion-based methods, and propagation-based methods. Each has its own advantages and is suitable for different scenarios. We chose to use saliency scores because they combine gradients and attention values, allowing us to effectively analyze the magnitude and trends of information flow between different components. Intuitively, the saliency score is only large when both the attention and gradient are large. This allows saliency to simultaneously reflect the relationships among various modules and their impact on the output, making it especially suitable for scenarios like ours, where multiple components are involved.

---

> ### Author Response · Authors · 2024-11-24
> **Inquiry Regarding Rebuttal Feedback**
>
> Dear Reviewer Y54T,
>
> We sincerely appreciate your time and effort in reviewing our manuscript and offering valuable suggestions. We provided detailed clarifications in response to your questions a few days ago. If you have any additional feedback, concerns, or questions regarding our response, we would greatly appreciate hearing from you and welcome further discussion.

---

> > ### Comment · Reviewer_Y54T · 2024-11-25
> >
> > I appreciate the author's response and the author effectively resolved my concerns. I still think it's a good paper and I keep my score.

---

> > > ### Author Response · Authors · 2024-11-25
> > >
> > > Thanks once again for your valuable comments. We are delighted to hear that your concerns have been resolved, and we are very grateful for your consistently positive evaluation of our work.

---

### Official Review · Reviewer_mj1a · 2024-11-07

**Soundness:** 2
**Presentation:** 3
**Contribution:** 2
**Rating:** 3
**Confidence:** 3

**Summary:**

This paper aims to improve prompt tuning on LLMs. The authors claim that later reasoning steps in wrong examples usually have high attention values on the soft prompt tokens, which is interpreted as "attend to the hints in soft prompt and conflict with early reasoning steps". Therefore, the authors propose to corrupt the soft prompt with a mask when such bad attention behavior is detected. Experiments on several math dataset show that this method can improve the vanilla prompt tuning.

**Strengths:**

- Several math benchmarks are covered in the experiment section, including various difficulty levels easy (gsm8k) and hard (MATH).
- The paper is well written and easy to follow.

**Weaknesses:**

- In this paper, soft prompt is interpreted as "carrier of task-related knowledge" or "hints", which is a bit casual and not well supported.

-  The major argument of this paper is LLM (later stage reasoning steps) shouldn't pay too much attention on soft prompts, which is very suspicious. Specifically,

     1. The authors derive this argument from empirical observation instead of rigorous causal relation analysis, or fundamental justification.
     2. The prompt / input to LLM, prompt is structured as {soft prompt; question} where soft prompt is in the leading position. Then the authors observe some high attention on soft prompt (leading tokens) on wrong examples. However, according to [1], LLM trends to put non-useful attention values on leading tokens (e.g., first several tokens) and that's why leading tokens has high attention value. In other words, it does not mean high attention on soft prompts, which are in leading position, is a bad behavior.
     3. On top of 2, the corruption mask that can improve reasoning accuracy is not because they mitigate bad high attention values on soft prompt, but something else, e.g., overfitting.

- In experiments table1, it's hard to validate any this prompt tuning methods (prompt tuning, act, dpc) is necessary or not on reasoning tasks. The authors should add full finetuning as the major baseline, and demonstrate that dpc is on par or even better than it.

[1]   Xiao, Guangxuan, et al. "Efficient streaming language models with attention sinks." arXiv preprint arXiv:2309.17453 (2023).

**Questions:**

Please find my comments in the weaknesses section

---

> ### Author Response · Authors · 2024-11-21
> **Response to Reviewer mj1a (1/3)**
>
> Thanks a lot for the time and effort you invested in providing the detailed reviews. Regarding the current weaknesses and questions you pointed out, we are glad to give our responses.
>
>
>
> **Weakness 1:**
> In this paper, soft prompt is interpreted as "carrier of task-related knowledge" or "hints", which is a bit casual and not well supported.
>
> **Response to Weakness 1:**
>
> Your comment is very insightful. Our interpretation is indeed inspired by the analysis in the key work on Prompt Tuning [1] [2].
>
> Specifically, section 7 of [1] mentions that "they observe that for a given learned prompt token, the top-5 nearest neighbors form tight semantic clusters. ... This suggests that the model is learning to store the expected output classes in the prompts **as reference**, and initializing the prompt to output classes makes this easier and more centralized." Additionally, it notes that "this suggests that one role of the prompt may be to prime the model to interpret inputs **in a specific domain or context**." ;
>
> Abstract of [2] mentions that "This suggests that while techniques like prompting, in-context learning, soft prompting, and prefix tuning can **effectively elicit skills present in the pretrained model**". This further supporting the idea that prompts serve as a means to activate and guide the model's latent knowledge.
>
> These findings provide a foundation for our interpretation of soft prompts as carriers of task-related knowledge, and we have already incorporated these references into the paper to clarify the origins and support for our perspective in the revised version.
>
>
>
> [1] Lester, Brian, Rami Al-Rfou, and Noah Constant. "The Power of Scale for Parameter-Efficient Prompt Tuning." Proceedings of the 2021 Conference on Empirical Methods in Natural Language Processing. 2021.
>
> [2] Petrov, Aleksandar, Philip Torr, and Adel Bibi. "When Do Prompting and Prefix-Tuning Work? A Theory of Capabilities and Limitations." ICLR2024
>
>
>
> **Weakness 2:**
> The authors derive this argument from empirical observation instead of rigorous causal relation analysis, or fundamental justification.
>
> **Response to Weakness 2:**
>
> Our "observation → hypothesis → validation" analysis method is inspired by some impactful works in the field, such as label words [1] and ACT [2]. This is a popular analytical paradigm in the domain. Meanwhile, as highlighted by Reviewer Y54T, we adopt a probing approach, quantitatively analyzing the efficacy of soft prompts in guiding complex reasoning within LLMs. We performed thorough qualitative and quantitative analyses to support our arguments, particularly through a step-by-step correlation analysis. The statistical correlation analysis can be considered as a rigorous justification.
>
> We begin with case-based observations of information flow within the model, as illustrated in Figure 3. Based on these findings, we propose a hypothesis: information accumulation in the shallow layers may correspond with  information flow pattern changes  in deep layers, and incorrect deep-layer flow patterns could lead to incorrect answers. This hypothesis is then substantiated through quantitative experiments. Figure 4(a) demonstrates a strong correlation between information accumulation and changes in deep-layer information flow patterns. Figures 4(b) and 4(c) further quantify the relationship between incorrect flow patterns and incorrect results by calculating  intensity of information flow  for the latter stages of reasoning in deep layers.
>
> Moreover, we employed a two-stage approach and conducted extensive experiments. The results showed consistent improvements, indicating that eliminating or reducing this phenomenon can enhance reasoning capabilities, thereby further validating our viewpoint.
>
> Ultimately, our findings confirm that soft prompts can guide models toward correct answers when later reasoning steps emphasize former tokens  more than soft tokens. Conversely, when soft prompts overly influence later reasoning steps, the model is more likely to produce incorrect answers. Beyond performance improvements, our work also offers novel insights to the fields of prompt tuning and PEFT. We hope to inspire more researchers to focus on the information flow trends among different input components and to uncover possible inherent connections.
>
>
>
> [1] Wang, Lean, et al. "Label Words are Anchors: An Information Flow Perspective for Understanding In-Context Learning."  emnlp2023.
>
> [2] Yu Z, Wang Z, Fu Y, et al. Unveiling and harnessing hidden attention sinks: Enhancing large language models without training through attention calibration. ICML24

---

> ### Author Response · Authors · 2024-11-21
> **Response to Reviewer mj1a (2/3)**
>
> **Weakness 3:**
> The prompt / input to LLM, prompt is structured as {soft prompt; question} where soft prompt is in the leading position. Then the authors observe some high attention on soft prompt (leading tokens) on wrong examples. However, according to [1], LLM trends to put non-useful attention values on leading tokens (e.g., first several tokens) and that's why leading tokens has high attention value. In other words, it does not mean high attention on soft prompts, which are in leading position, is a bad behavior.
>
> **Response to Weakness 3:**
>
> Thank you for your insightful comments. To address your concerns, we have added visualizations of the attention matrix in Appendix C  and some related work comparison in the Appendix B of the revised version, please review. **Specifically, we also observed the Attention Sink phenomenon, this does not conflict with our method.** Next, I will explain the differences between each method in combination with the illustrations in Appendix B.
>
> Firstly, as shown in Appendix B Figure 6(a), our analysis tool differs from that used in StreamingLLM[1]. While StreamLLM uses attention scores, we utilize saliency scores, which combine gradients and attention values. This allows us to effectively analyze information flow between different components, making it suitable for scenarios like ours that involve multiple components. This methodological difference may lead to different analysis results.
>
> Additionally, as illustrated in Appendix B Figure 6(b), StreamingLLM observes high attention values on **the first token**, termed an "**attention sink**," and leverages this finding to extend the input sequence length—a positive use of high attention values.
>
> However, **different scenarios can exhibit different behaviors**. For example, as shown in Appendices B Figure 6(c), (d), and (e), several studies have demonstrated the **negative effects** **of the attention sink** phenomenon.
>
> The ACT[2] study found that attention sinks not only occur on the first token but also on **tokens with limited semantic information** (e.g., ".", ":", and "<0x0A>") , as visualized in Appendix B Figure 6(c). Contrary to the observations made in StreamingLLM—which suggest preserving attention sinks to enhance LLMs' accuracy—**they highlight that not all attention sinks are beneficial**. Specifically, for most attention sinks occurring in the middle or later parts of inputs, reducing their attention scores can lead to improved accuracy.
>
> In OPERA[3] and DOPRA[4], as represented in Appendix B Figure 6(d)(e),it was found that when models generate hallucinated content, the self-attention weights in the last layer exhibit a **distinct "columnar" pattern** before the **hallucination occurs**, leading to an "over-trust" tendency in the self-attention weights for the hallucinated parts.
>
> Although our soft prompt tokens precede the question tokens, our setup differs significantly from that of StreamingLLM. Our soft prompt is of considerable length,and the phenomenon of information accumulation often occurs not at the first token but in the middle portion. Furthermore, we do not assert that all instances of information accumulation have negative effects. In our method, we differentiate whether the information accumulation has negative impacts and have demonstrated this through ablation experiments.
>
>
>
> [1] Efficient Streaming Language Models with Attention Sinks. ICLR 2024
>
> [2] Unveiling and Harnessing Hidden Attention Sinks: Enhancing Large Language Models without Training through Attention Calibration. ICML 2024
>
> [3] OPERA: Alleviating Hallucination in Multi-Modal Large Language Models via Over-Trust Penalty and Retrospection-Allocation. CVPR 2024
>
> [4] DOPRA: Decoding Over-accumulation Penalization and Re-allocation in Specific Weighting Layer. ACM MM 2024
>
>
>
> **Weakness 4:**
> On top of 2, the corruption mask that can improve reasoning accuracy is not because they mitigate bad high attention values on soft prompt, but something else, e.g., overfitting.
>
> **Response to Weakness 4:**
>
> As outlined above, our approach does not conflict with Attention Sinks. Our approach employ the saliency score, not the attention score, to identify the soft prompts that require intervention, which does not conflict with the high attention weights of the first tokens due to the need for LLMs to release unnecessary attention. We have conducted both qualitative and quantitative analyses to support our findings.
>
> Furthermore, our ablation study includes a random mask experiment, which shows that applying random masks does not lead to consistent performance improvements. This further demonstrates that our improvements are not due to overfitting.

---

> ### Author Response · Authors · 2024-11-21
> **Response to Reviewer mj1a (3/3)**
>
> **Weakness 5:**
> In experiments table1, it's hard to validate any this prompt tuning methods (prompt tuning, act, dpc) is necessary or not on reasoning tasks. The authors should add full finetuning as the major baseline, and demonstrate that dpc is on par or even better than it.
>
> **Response to Weakness 5:**
>
> Prompt tuning is a form of parameter-efficient fine-tuning (PEFT), which includes notable approaches like LoRA, prefix tuning, and prompt tuning, each with its own advantages and application scenarios. The essence of PEFT lies in achieving performance improvements with significantly fewer resources compared to full fine-tuning.
>
> Following your suggestion, we conducted extensive experiments using prefix tuning, LoRA, and full fine-tuning. Our DPC method consistently demonstrated improvements within the scope of prompt tuning. Furthermore, our results show that DPC achieves optimal or near-optimal performance with relatively few computational resources compared to multiple methods, confirming its effectiveness. Notably, we observed that LoRA and full fine-tuning sometimes led to performance degradation. We believe this may be because modern LLMs have undergone comprehensive post-training processes, including Supervised Fine-Tuning (SFT) and Reinforcement Learning from Human Feedback (RLHF). Consequently, fine-tuning model parameters on a single dataset can be challenging and may even result in decreased performance. Therefore, DPC enhances performance while more easily avoiding the risk of performance degradation by not altering the fragile and easily disrupted model parameters.
>
>
>
> |                      |           | Llama2   |          |           | Llama3   |          |           | Mistral  |          |
> | -------------------- | --------- | -------- | -------- | --------- | -------- | -------- | --------- | -------- | -------- |
> | **Method/Datasets**  | **GSM8k** | **MATH** | **AQuA** | **GSM8k** | **MATH** | **AQuA** | **GSM8k** | **MATH** | **AQuA** |
> | **Pretrained model** | 29.5      | 2.0      | 21.0     | 64.9      | 30.0     | 34.0     | 37.9      | 5.1      | 26.0     |
> | **Prompt tuning**    | 38.1      | 7.6      | 22.4     | 65.5      | 33.7     | 38.5     | 49.5      | 15.0     | 28.7     |
> | **Prefix tuning**    | 41.7      | 8.4      | 20.1     | 65.4      | 33.0     | 41.3     | **54.4**  | 16.3     | 31.5     |
> | **LoRA**             | 12.7      | 7.4      | 24.8     | 44.1      | 31.8     | 38.6     | 44.9      | **19.9** | 24.4     |
> | **Full fine-tuning** | 5.2       | 8.2      | 24.8     | 40.9      | 27.1     | **42.9** | 45.1      | 12.4     | 26.4     |
> | **DPC**              | **41.9**  | **9.2**  | **31.1** | **67.6**  | **36.3** | *42.5*   | *51.1*    | *16.4*   | **31.9** |
>
> The **bolded numbers** are the best results, and the *italic numbers*  represent the second-best results.

---

> > ### Comment · Reviewer_mj1a · 2024-11-29
> >
> > Really appreciate the authors for the rebuttal and new results. I apologize for my late response.
> >
> > Weakness 1 and 2 are well addressed, many thanks!
> >
> > Regrading weakness 3 and 4, thanks for the detailed comparison. However, I don't think my concern was solved. Basically,  all the four papers the authors attached agree that high attention are not always good. According to [1], since the softmax normalizes the attentions into 1, it is inevitable to have high values, therefore model will find places to store this high garbage values. While [2] find tokens with less semantic meanings are used, both [1][2] find initial tokens are used to store these high values h (i.e., attention sink). I understand the authors use saliency instead of attention, but my question was adding soft prompt before the questions will make these soft prompt tokens to serve the attention sink purpose. That is to say, using mask on them will disturb it. Therefore, it is unclear this masks help model in general, or the improvement is just from something else, e.g., overfitting to the benchmark.
> >
> > Regrading weakness 5, I think PEFT makes sense if it can achieve at least comparable results to full finetuning. Really thank the authors for the additional results. However, i notice full fine-tuning get 29.2->5.2 (llama2 GSM8K), 64.9->40.9 (llama3 GSM8K), 30->27.1 on (llama3 MATH). While full-finetuning gets much worse results? This seems to contradicts with many previous findings.
> >
> > Looking forward to more response from the authors.

---

> > > ### Author Response · Authors · 2024-11-29
> > > **Further Response to Reviewer mj1a (1/2)**
> > >
> > > Dear Reviewer mj1a,
> > >
> > > Thank you very much for your thoughtful response. We greatly appreciate your insightful comments. In our previous rebuttal, we mentioned these phenomena but did not provide a comprehensive explanation. We are very grateful for the opportunity to offer a more detailed clarification here.
> > >
> > > **Regarding weaknesses 3 and 4:**
> > >
> > > **1. soft prompt tokens serve the attention sink purpose**
> > >
> > > Firstly, we agree with your insightful observation that placing soft prompts before the question leads to attention sink behavior. As shown in Appendix C, Figure 7 of the current manuscript, we visualized the attention scores and confirmed that the initial tokens of the soft prompt exhibit the same phenomenon as in [1].
> > >
> > > However, a key point we would like to clarify is that attention sinks primarily appear at **the initial tokens**, both in [1] (where attention sinks are observed at the first token(Figure 2 of [1]), and the author just used the **4 initial tokens**) and in our soft prompt setup. Indeed, our soft prompt is of considerable length, consisting of a total of **100 tokens**, and *while the attention sinks appear at the initial tokens (i.e., the first few tokens), this does not necessarily align with the position of the highest saliency scores (i.e., information accumulation), which typically occur in the middle part of the soft prompts*. As illustrated in Figure 2(b), which visualizes the saliency scores, the initial tokens have low saliency scores, while those in the middle positions have much higher scores. Therefore, due to the differences between the mechanisms of attention and saliency, the location of the attention sinks does not correspond to the final position we select for corruption. Thus, these two phenomena are not in conflict; we have not interfered with the initial tokens that serve as attention sinks.
> > >
> > > Our experimental analysis in section 2.2.1, which examined the relationship between information accumulation and changes in deep-layer information flow patterns, demonstrated a strong correlation between high saliency scores (i.e., information accumulation) and erroneous information flow patterns. This insight inspired the development of our DPC method. Specifically, by identifying erroneous information flow patterns in deeper layers, we were able to pinpoint corruption locations based on the observation of information accumulation.
> > >
> > > [1] Xiao, Guangxuan, et al. "Efficient streaming language models with attention sinks." arXiv preprint arXiv:2309.17453 (2023).
> > >
> > > **2. about overfitting**
> > >
> > > Firstly, in our ablation studies, we conducted random corruption experiments, which included stochastically selecting mask positions and applying mask randomly. Neither of these setups led to consistent performance improvements, indicating that random masks has minimal impact. This supports the view that only precisely executed corruption, based on the identified associations, can lead to meaningful improvements. These empirical results demonstrate that our improvements are not due to overfitting.
> > >
> > > Additionally,  Reviewer sfZi mentioned the issue of generalization in his comments. We have included the experimental results here for reference. Specifically, we conducted further experiments on another three datasets involving logical reasoning and commonsense reasoning, which demonstrated that DPC not only excels in the initial tasks but also effectively generalizes to other reasoning scenarios. This observed generalization across different datasets demonstrates the robustness of DPC and further supports that our improvements are not due to overfitting.
> > >
> > >
> > > |                     |              | Llama3-8b         |               |              | Mistral-7b        |               |
> > > | ------------------- | ------------ | ----------------- | ------------- | ------------ | ----------------- | ------------- |
> > > | **Method/Datasets** | **prontoQA** | **ARC-challenge** | **WorldTree** | **prontoQA** | **ARC-challenge** | **WorldTree** |
> > > | **Prompt tuning**   | 100          | 13.2              | 39.1          | 97.9         | 14                | 55.2          |
> > > | **DPC**             | **100**      | **14.8**          | **45.5**      | **98.9**     | **16.6**          | **56.5**      |
> > >
> > > [1] Xiao, Guangxuan, et al. "Efficient streaming language models with attention sinks." arXiv preprint arXiv:2309.17453 (2023).

---

> ### Author Response · Authors · 2024-11-24
> **Inquiry Regarding Rebuttal Feedback**
>
> Dear Reviewer mj1a,
>
> We sincerely appreciate your time and effort in reviewing our manuscript and offering valuable suggestions. We provided detailed clarifications in response to your questions a few days ago. If you have any additional feedback, concerns, or questions regarding our response, we would greatly appreciate hearing from you and welcome further discussion.

---

> ### Author Response · Authors · 2024-11-29
> **Further Response to Reviewer mj1a (2/2)**
>
> **Regarding weaknesses 5:**
>
> **1. Why do full fine-tuning methods sometimes perform worse?**
>
> We mentioned this phenomenon in our previous rebuttal, and here we provide a more thorough explanation.
>
> All of our experiments were conducted based on the Instruct models, and our SFT setting is quite simple. Specifically, our hyper-parameters were as follows: learning rate = 1e-5, weight decay = 0.1, cosine learning rate scheduler, max gradient norm = 1.0, warm-up ratio = 0.03, batch size = 4, max length = 2048, *direct QA format training and a single dataset*. We speculate that the decline in SFT performance might due to the simplicity of our SFT setup, while the modern base models have already been powerful. The reasons are below.
>
> In fact, the LLaMA2 / LLaMA3 Instruct models we used in this paper had already been fine-tuned on various datasets (including *mathematical datasets*) [1] during their post-training phase to enhance instruction-following and reasoning capabilities across a wide range of downstream tasks. Therefore, we speculate the performance degradation in our SFT experiments occurred because our relatively simple training setup (e.g., *single dataset, direct QA format*, etc.) disrupted the models' original instruction-following abilities. This issue can be mitigated by mixing multiple datasets, particularly by incorporating instruction-following datasets and modifying the QA training format(e.g., add some instructions to the question)[2]. However, such optimizations fall under the scope of SFT performance tuning and are not part of our proposed method. We will clarify the specific experimental settings in the tables of our revised manuscript to avoid confusion.
>
> To summarize, our SFT experiments serve as a baseline comparison. Despite using a relatively simple experimental setup, *SFT still performs well in certain cases*. Additionally, in some scenarios, PEFT methods also yield good performance. Therefore, we recommend trying out a combination of different post-training techniques, such as Prompt Tuning, LoRA, and SFT, to adaptively achieve better results in different settings and scenarios.
>
>
> **2. The Value of DPC**
>
> As we have described earlier, different methods such as PEFT and SFT can be combined and experimented with for a specific scenario. In particular, for resource-constrained environments, soft prompts offer a significant advantage: compared to full SFT, they can achieve comparable performance while fine-tuning only about 0.01% of the model's parameters scale. Furthermore, soft prompts serve as a plug-in module to the backbone model, preserving the core capabilities of backbone model, and makes them an excellent choice for low-budget scenarios. Our DPC approach, grounded in the field of prompt tuning, provides an efficient solution for performance enhancement.
>
>
> [1] Instruction Tuning for Large Language Models: A Survey. arXiv:2308.10792
>
> [2] Dynamics of Instruction Tuning: Each Ability of Large Language Models Has Its Own Growth Pace. arXiv:2310.19651

---

### Public Comment · ~Jiahao_Zhao1 · 2025-04-08
**Great Work! Any plan to release a repo?**

This work is both innovative and solid. Would there be any plans to release a full repository? Access to the complete code would greatly benefit the research community and allow for further exploration of your promising approach.

Looking forward to hearing from you.

---

> ### Public Comment · ~Chen_Shen7 · 2025-04-18
>
> Thanks for your advice. I will let you know when the code is open source: )

---

### Meta-Review · Area_Chair_DAxn · 2024-12-20

**Metareview:**

The paper investigates when and how the soft prompt might favor or adversely affect an LLM reasoning performance. It shows empirically that "soft prompt tokens in certain positions have a strong impact on reasoning"; and it proposes the so-called dynamic prompt corruption approach, firstly inspecting the impact factor of the soft prompt on the latter part of the reasoning in the final layer, and secondly masking the key tokens that most influence both the question and reasoning steps in shallow layers.

**Additional Comments On Reviewer Discussion:**

The rebuttal did a good job in addressing most reviewers' concerns, notably providing complementary experiments, or explaining why the saliency indicator, measuring the joint intensity of the attention and gradient.
Discussing and clarifying the relation between attention sinks and the emphasis put on soft tokens would be appreciated in the final version.

---

### Decision · Program_Chairs · 2025-01-22

Accept (Poster)